# Conflict driven displacement and child health: Evidence based on mother's nationality from Jordan Population and Family Health Survey

Manzoor Ahmad Malik[1], Saddaf Naaz Akhtar[2]*, Rania Ali Albsoul[3], Muhammad Ahmed Alshyyab[4]

**1** Department of Humanities and Social Sciences, Indian Institute of Technology (IIT), Roorkee, India, **2** Centre for the Study of Regional Development, School of Social Sciences-III, Jawaharlal Nehru University, New Delhi, India, **3** Department of Family and Community Medicine, The University of Jordan, Amman, Jordan, **4** Faculty of Medicine, Department of Public Health and Community Medicine, Jordan University of Science and Technology, Irbid, Jordan

* sadafdpsjsr@gmail.com

## Abstract

### Introduction

Armed conflicts result in greater vulnerability and socioeconomic inequality of populations besides risking their health and well-being. Conflict intensifies the health needs and risks the life and well-being of individuals at large through displacement. Therefore, our study aims to apprise the interventions to which children under-five living in Jordan are especially at risk for acute respiratory infections, diarrhea, and fever in the conflict circumstances.

### Materials and methods

Secondary data analysis is used in the present study. We used a weighted sample of around 9650 children from Jordan Population and Family Health Survey (JPFHS), 2017–18. Bivariate analysis including prevalence rates were used to examine the distribution of socio-demographic characteristics of children. The study has also used multinomial logistic regression model, in order to evaluate the variations in the probability of nationality of under-five children living in Jordan.

### Results

"Syrian nationalist" children have a higher relative risk of ARI (RRR = 1.19, [1.08, 1.32]), and "Other-nationalist" children have two times greater risk of ARI compared to "Jordanian children." The relative risk of diarrhea is lower among "Syrian nationalist" and "Other-nationalist" children compared to "Jordanian children." Children belong "Other-nationalist" are found to be less relative risk of fever (RRR = 0.9, [0.80, 1.01]) than "Jordanian children."

### Conclusions

Our study concludes that conflict-driven displacement has an immediate effect on child health through access, availability, and affordability of health care services, resulting in

**Data Availability Statement:** The data analyzed are publicly available. https://dhsprogram.com/what-we-do/survey/survey-display-500.cfm.

**Funding:** The author(s) received no specific funding for this work.

**Competing interests:** The authors have declared that no competing interests exist.

**Abbreviations:** ARI, Acute respiratory infections; BCG, Bacille Calmette-Guerin; DHS, Demographic and health surveys; GDP, Gross domestic product; JPFHS, Jordan Population and Family Health Survey; RRR, Relative risk ratio; UNHCR, United Nations High Commissioner for Refugees.

more significant health care risks. However, sufficient investment is required to address such adversities that affect the health care system due to uneven demand as experienced by the Jordanian health care system. Thus, collaborative efforts through global partners can play a significant role in the countries facing the challenges of managing these health care emergencies.

## Introduction

Armed conflicts result in greater vulnerability and socioeconomic inequality of populations besides risking their health and well-being. The inevitable impact of armed conflict results in severe public health challenges, mediated by displacement and collapse the health care system [1,2]. Throughout the Conflict impacts socioeconomic and health care outcomes through direct and indirect effects, like violence, injuries, forced labor, exploitation, and unlawful detention [3]. Due to conflict-driven displacement, populations at risk face inadequate health services apart from losing their livelihood and facing the challenges of socioeconomic marginalization [4]. Even though removal lowers their risk of death due to violence, the lack of health care services and proper attention from health care service providers marks them uncovered against health risks [5]. The challenges further evolve through availability, affordability, and accessibility of these health care services in the displaced locations [6,7].

Conflict intensifies the health needs and risks the life and well-being of individuals at large through displacement. Populations displaced to other countries, face severe challenges of inadequate health care due to the loss of their livelihood and shelter [8]. Conflict forces them to displace and take refuge in the areas, where they mostly remain concentrated in refugee camps and unmarked regions [9]. These places are already fragile, and the lack of basic amenities keeps them exposed to socioeconomic and health care vulnerabilities [10]. One of the worst challenges they face is acceptance from the host country, which further deteriorates their well-being [11,12] Other possible challenges evolve, through unexpected demand, lack of facilities in the host country, and the kind of health care provision [13]. Conflict further increases the risk of nutritional challenges, mental health, sexual abuse, and other health-related challenges that likely increase the disease burden [14–16]. Thus, conflicts driven displacement profoundly impacts the population's health and well-being, especially vulnerable groups like women, children, and the elderly.

Children are the most susceptible group that suffers from any calamity, be it natural or human made. Children face severe consequences due to conflict, ranging from their health, education, and material welfare. Conflict-driven displacement impact children in terms of diseases and loss of life that are mostly avoidable [17]. With 250 million children live in the world's most vulnerable zones at risk of both life loss and basic health care services [18]. Children are directly exposed to health risks through conflict with under-five aged at most [19]. Children suffer both directly and indirectly to the effects of conflict than the other vulnerable groups like women and elderly [20]. Conflict affects children through preventable diseases, lack of basic necessities, sexual abuse and proper family care [21]. Conflict driven displacement further enhances the greater inequality of health among children and their physical growth [22]. Previous studies have found that children, especially at their early age, are highly vulnerable to various morbidity outcomes, malnutrition, and mental health challenges, and these outcomes further complicate when they face the conflict-driven displacement and migrant atmosphere [23–25].

## Theoretical construct

Syria is one of the worst affected countries in recent times due to conflict. According to estimates, around 8.8 million people were displaced due to the ongoing conflict, with one-quarter of these people entering into their neighbouring countries through international borders [26]. One of the most affected populations of these displaced people is under-five children. According to estimates, around 48 percent of the displaced people are under 18 years of age, and approximately five in every registered refugee are children of under-five age [26]. The Fig 1 below shows the estimates of people displaced due to conflict in Syria.

Owing to the fact that the conflict in the region remains pounding. Nearly 4 out of every 5 refugees lives in countries neighboring their countries of origin [26]. Since the conflict began in Syria in 2011, Jordan has shouldered the impact of a massive influx of Syrian refugees, and currently, Syrian refugees constitute around 10 percent of Jordan's population, which has placed immense pressure on the country's over-stretched resources at one of the most difficult economic periods in its history [27]. Nearly 78,000 Syrian refugees reside in rows of assembled shelters in Jordan [27]. Furthermore, Jordan has the second-highest share of refugees per capita of GDP [28], which puts immense pressure on the Jordanian health care system to suffice the displaced people's needs apart from its population. Since the displaced people are prone to a lack of financial resources and mainly depend on out-of-pocket expenditures to bear the health care costs. This has exerted extra pressure on Jordanian health care systems to meet the unprecedented demand and provide adequate health resources to these displaced populations [29]. Furthermore, since 70 percent of refugees reside in Jordanian communities, this makes it difficult for the health care system to differentiate or exclude the refugees [30]. Although Jordan is known for its exceptional health care system, with higher GDP spending of around 7.2 percent on health care [31]. It is one of the most updated health care systems in the Arab world functioning through multiple organs, including public and private institutions. [32,33]. But the increasing political instability and cross-border conflict have affected it on a grander scale [34].

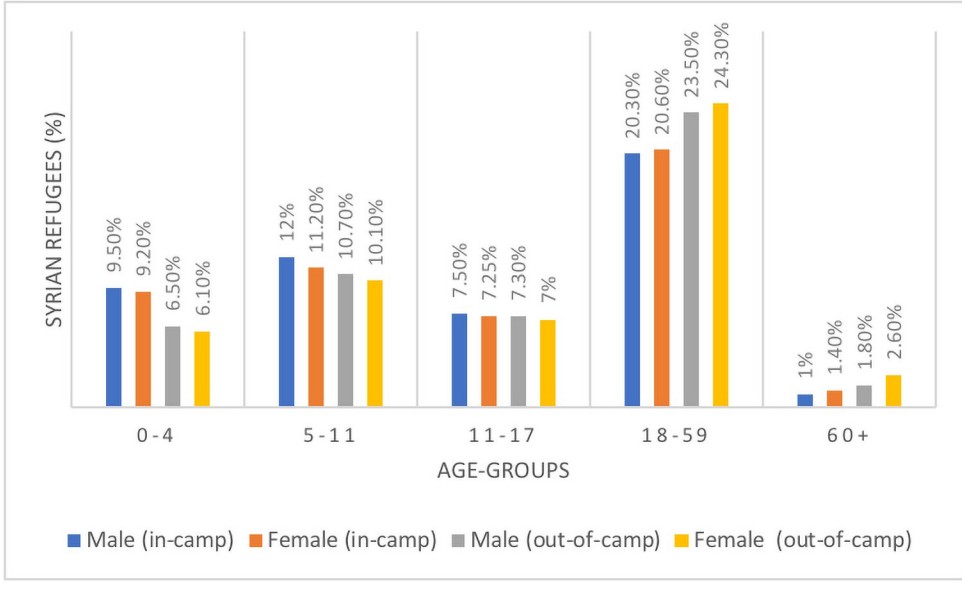

**Fig 1. Percentage of Syrian refugees by age-groups and sex in Jordan by September 2020.** Source: United Nations High Commissioner for Refugees (UNHCR).

Jordan is politically vulnerable due to its resource impoverishment, externally oriented rentier economy, limited internal revenue sources, and extreme population growth [35]. Syrian conflict, Jordan has become increasingly wary of its growing Syrian populations [35]. Jordan provides sufficient health care access to people from Syria and other countries [36], but despite that, there is a gap in accessing healthcare services among many Syrian refugees in Jordan due to the barriers, such as not possessing the civil documentation or having the means to pay out-of-pocket expenditures for accessing adequate health care services [29]. With already poor living conditions, the Syrians are thus more susceptible to disease outbreaks and suffer from a lack of health care services [36]. Displacement through conflict is an increasing challenge and puts large populations at risk, especially children since conflict-driven displacement has long and short-term health effects [37]. It also affects the host country's health care system with exerted demand due to the influx of displaced populations. Therefore, given the challenge, it is key to amplify the policy gaps and reflect the impact of conflict on health care outcomes. This study further aims to apprise interventions to which children under five are especially at risk for acute respiratory infections, diarrhea, and fever in the conflict circumstances. Moreover, there is a dearth of research in this context in the literature, which can reflect the linkages and examine children's health care scenarios due to conflict-driven displacement, especially in the Arab world. Thus, based on the above theoretical connotation, we used the JPFHS data on children and identified them based on mother's nationality to examine the child health inequality among the children, particularly those displaced from the neighboring countries, especially Syria in Jordan, due to conflict.

## Materials and methods

### Data source and survey description

This study is based on demographic health survey of Jordan. We used the Jordan Population and Family Health Survey (JPFHS) which has been implemented by the Department of Statistics (DOS) from early October 2017 to January 2018 [38]. The JPFHS (2017–18) is the seventh to be conducted in Jordan and follows the 1990, 1997, 2002, 2007, 2009, and 2012 JPFHS surveys. The survey provides valuable information on trends in key demographic and health indicators over time. The collected information is intended to assist policymakers and program managers in evaluating and designing programs and strategies for improving the health of the country's population. Additionally, for the first time in Jordan, the 2017–18 JPFHS included a male survey. The survey collected information on women's basic demographic and social characteristics, on their knowledge and use of family planning methods, and on their knowledge and attitudes towards HIV and other sexually transmitted infections [38]. For conducting this study, we used a weighted sample of around 9650 children. As this study utilized secondary data from the JPFHS, it was exempted from the institutional review board evaluation.

**Outcome variable.** The main dependent variable in the study is nationality of mother. This is the only variable that can provide information about the refugees in the survey. Although it may not be the case that every mother with other nationality is a refugee in Jordan, but based on previous literature [39–41], it is clear that most mothers with other and Syrian nationality are refugees settling in Jordan due to the ongoing conflict in Syria and neighboring countries of Jordan. Therefore, we used the mother's nationality to differentiate the health outcomes of children belonging to mothers with a different nationality. In questionnaire it has been asked to the respondent that" What is your nationality?" The nationality of mother has been categorized into three categories: Jordanian, Syrian and other-nationalists (includes Egyptian, Iraqi, other Arab nationalities and non-Arab nationalities).

**Explanatory variables.** Socio-economic factors and demographics factors are: Age of child, sex of child, place of residence, regions, wealth index, mothers' education, fathers' education, fathers' working status, mothers' working status, birth order, number of living children, health services factors (like health card and received BCG vaccine), wash and hygiene factors (like-sources of water supply, clean water, toilet facilities), environmental factors (like fuel type, mother's smoking cigarette, mothers' smoking water).

**Data analysis.** Bivariate analysis including prevalence rates were used to examine the distribution of socio-demographic characteristics of children. The study explores the different morbidity outcomes of children like acute respiratory infections (ARI), diarrhea and fever by three nationalities. The independent variables include ARI fever and Diarrhea apart from the other socio-economic and demographic characteristics of children and mother.

The study has used multinomial logistic regression model to evaluate the health risks based on the mother's nationality of under-five children living in Jordan. The relative risk ratios have been determined by using multinomial logistic regression for the given independent variables for each category of the nationality of under-five children living in Jordan (dependent variable). The regression model is then fitted to examine the association between a set of independent variables such as prevalence of ARI, diarrhea, fever, and other socio-economic variables (Mothers' education, wealth index, place of residence and region, explaining the likelihood of the Syrian nationality of under-five children living in Jordan. The equation fitted to the data are as follows:

$$\ln \frac{P(y_i = m)}{P(y_i = l)} = \alpha + \sum_{k=1}^{K} \beta_{xm} x_{ik} = z_{mi}$$

The dependent variable has been categorized into three categories (Jordanian, Syrian and Other-nationalists) and represented by $m$ as mentioned in the above equation. The main reason of selecting this as a dependent variable was to analyze the difference in health outcomes of the children belonging to Syria, Jordan and other nations. Hence, there is need to calculate for $m - 1$, one for each category relative to the reference category in order to illustrate the probability of the prevalence of ARI, diarrhea & fever, and other socio-economic variables (Mothers' education, wealth index, place of residence and regions). For the Others nationality category of the dependent variable, the derived equation is then estimated are given as follows:

$$P(y_i = m) = \frac{\exp(z_{mi})}{1 + \sum_{h=2}^{M} \exp(z_{hi})}$$

The Jordanian nationality of under-five children living in Jordan has been selected as the comparison category. The model parameter estimates and the Relative Risk Ratio (RRR) for the multinomial logit model is that for a unit change in the predictor variable, the logit of outcome m relative to the reference group is expected to change by its respective parameter estimate given that the variables in the model are held constant [42,43]. All analyses were conducted using STATA 14.0.

# Results

Table 1 presents the percentage of under-five children living in Jordan with different nationalities by background characteristics (2017–18). The background characteristics include the age of under-five children (in months), sex, place of residence, regions, wealth index, mother's education, father's education, father's working status, mother's working status, birth order, and the number of children living.

**Table 1. Background characteristics by nationalities among under-five children in Jordan, 2017–18.**

| Background | Jordanian | Syrian | Others |
|---|---|---|---|
| **Age** | | | |
| Less than 6 months | 84.67 | 11.79 | 3.53 |
| 6–11 months | 85.61 | 11.79 | 2.61 |
| 12–23 months | 83.42 | 13.02 | 3.56 |
| 24–35 months | 84 | 13.24 | 2.76 |
| 36–47 months | 83.18 | 11.55 | 5.27 |
| 48–60 months | 83.23 | 12.32 | 4.44 |
| **Sex** | | | |
| Male | 83.74 | 12.12 | 4.14 |
| Female | 83.88 | 12.64 | 3.48 |
| **Place of Residence** | | | |
| Urban | 82.33 | 13.52 | 4.14 |
| Rural | 95.11 | 3.56 | 1.33 |
| **Regions** | | | |
| North | 78.94 | 18.85 | 2.21 |
| Central | 84.82 | 10.03 | 5.15 |
| South | 94.99 | 3.6 | 1.41 |
| **Wealth Index** | | | |
| Poorest | 62.9 | 32.84 | 4.26 |
| Poorer | 86.36 | 9.37 | 4.27 |
| Middle | 94.09 | 3.65 | 2.25 |
| Richer | 95.16 | 2.3 | 2.54 |
| Richest | 91.25 | 1.65 | 7.1 |
| **Mothers' Education** | | | |
| No Education | 49.3 | 38.62 | 12.09 |
| Primary | 37.48 | 53.6 | 8.91 |
| Secondary | 83.12 | 13.56 | 3.32 |
| Higher | 94.7 | 2.08 | 3.22 |
| **Fathers' Education** | | | |
| No Education | 67.14 | 26.55 | 6.3 |
| Primary | 53.59 | 42.51 | 3.9 |
| Secondary | 85.93 | 10.98 | 3.09 |
| Higher | 90.8 | 4.01 | 5.2 |
| Don't know | 93.63 | 6.37 | 0 |
| **Fathers' Working Status** | | | |
| Not Working | 71.38 | 22.54 | 6.08 |
| Working | 85.82 | 10.73 | 3.45 |
| **Mothers' Working Status** | | | |
| Not Working | 82.16 | 13.9 | 3.94 |
| Working | 95.22 | 1.77 | 3.01 |
| **Birth Order** | | | |
| <2 | 84.39 | 11.11 | 4.5 |
| 3–5 | 84.9 | 11.78 | 3.32 |
| 6+ | 75.84 | 21.43 | 2.73 |
| **Number of children living** | | | |
| Single | 84.82 | 9.75 | 5.43 |
| Three | 85.2 | 11.29 | 3.5 |
| More than three | 81.62 | 14.79 | 3.58 |

Source: Authors' estimation using JPFHS (2017–18).

**Table 2. Sample size distribution of acute respiratory infection, diarrhea and fever by nationalities among under-five children in Jordan, 2017–18.**

| Nationality | ARI | Diarrhea | Fever | Total Sample[a] |
|---|---|---|---|---|
| Jordanian | 5.76 | 9.8 | 13.01 | 8065 |
| Syrian | 6.45 | 8.76 | 13.57 | 1191 |
| Other | 10.13 | 8.49 | 13.01 | 368 |
| Total | 6.02 | 9.62 | 13.09 | 9623 |

[a]Unweighted total sample of Children by mother's nationality.

Source: Authors' estimation using JPFHS (2017–18).

Table 2 presents the percentage and sample distribution of ARI, diarrhea, and fever among under-five children by nationalities in JPFHS (2017–18). The Jordanian nationality constitutes 8065 under-five children; Syrian nationality includes 1191 under-five children and Other-nationality 368 under-five children. The percentage of ARI and fever among other nationalist and Syrian under-five children are found to be higher than Jordanian under-five children. While in the case of Diarrhea, Jordanian under-five children are found to be 1% greater than Syrian under-five children.

Table 3 presents the prevalence of acute respiratory infection, diarrhea, and fever by socio-economic characteristics among under-five children in Jordan, 2017–18. Our study found that the prevalence rate of ARI among Other-nationalist (9.56%) and Syrian (6.67%) under-five children are found to be higher than Jordanian (5.69%) children in urban areas. The prevalence rate of ARI among all nationalist's under-five children in rural areas are seen to be lower than urban children. Mother with no education has the lowest prevalence rate of ARI among Jordanians (1.17%) and Syrian (1.88%). Mother with higher education has high prevalence rates among Other-nationalist (13.25%) and Syrian (7.25%). Wealthier children have the highest ARI prevalence rate, with 19.92% among other-nationalists than Jordanian and Syrian. Syrian and Other-nationalist female children have a greater prevalence of ARI than male children, but in the case of Jordanian male has a higher prevalence rate of ARI than female. Syrian and Jordanian children aged 6–11 months are positively affected with ARI, but Other-nationalist children aged 12–23 months have the highest prevalence rate. Jordanian and Syrian children whose father and mother are working are positively affected by ARI than non-working parents. The lowest prevalence is observed in the North Region, where Jordanian children have 4.86%, Syrian children have 4.79%, and Other-nationalist has 5.41%. In contrast, the highest observed in the central part for Jordanian and Syrian but in the South region, Other-nationalist.

Similarly, the prevalence of diarrhea is higher among urban Jordanian and Syrian than rural, but children who belong to rural Other-nationalist have higher than urban. Mother with secondary education has a higher prevalence rate among Other-nationalist (13.57%). Jordanian children are highly infected with diarrhea than Syrian children concerning their mother's education level. Simultaneously, female children are found to have greater diarrhea prevalence among all nationalists than male children. Syrian children age less than six months have the highest prevalence rate, with 22.75%. Jordanian children whose father is not working are observed to have the highest prevalence rate of 11.34%, while the Syrian working mothers have the highest prevalence rate, with 14.68%. Syrian and Other-nationalist children belong to the southern region are highly infected with diarrhea.

Despite that, rural Jordanian and 'Other-nationalist' children have a high prevalence of fever than urban children. Still, urban Syrian children are more likely to have a higher prevalence of fever than rural. Children whose mothers have secondary education levels belong to

**Table 3. Prevalence of acute respiratory infection, diarrhea and fever by socioeconomic characteristics among under-five children in Jordan, 2017–18.**

| Socioeconomic characteristics | ARI | | | Diarrhea | | | Fever | | |
|---|---|---|---|---|---|---|---|---|---|
| | Jordanian | Syrian | Others | Jordanian | Syrian | Others | Jordanian | Syrian | Others |
| **Place of Residence** | | | | | | | | | |
| Urban | 5.69 | 6.67 | 9.56 | 9.81 | 8.76 | 8.06 | 12.72 | 13.59 | 13.34 |
| Rural | 6.26 | 0 | 23.57 | 9.73 | 8.56 | 18.53 | 14.86 | 13.2 | 21.02 |
| **Mothers' Education** | | | | | | | | | |
| No Education | 1.17 | 1.88 | 7.12 | 5.75 | 4.08 | 0 | 4.64 | 7.2 | 11.32 |
| Primary | 5.94 | 6.43 | 1.89 | 11.3 | 8.65 | 3.55 | 11.14 | 13.94 | 13.93 |
| Secondary | 6.11 | 6.77 | 11.26 | 10.04 | 9.56 | 13.57 | 14.17 | 13.81 | 15.72 |
| Higher | 5.44 | 7.25 | 13.25 | 9.49 | 5.93 | 5.26 | 11.92 | 14.77 | 11.08 |
| **Wealth Index** | | | | | | | | | |
| Poorest | 5.76 | 5.47 | 15.72 | 11.91 | 8.46 | 7.69 | 13.79 | 12.99 | 10 |
| Poorer | 6.86 | 8.9 | 7.4 | 9.31 | 8.17 | 16.01 | 14.84 | 17.52 | 22.14 |
| Middle | 5.05 | 15.9 | 0 | 9.51 | 21.21 | 2.27 | 13.82 | 12.66 | 11.64 |
| Richer | 5.63 | 0 | 19.92 | 8.78 | 0 | 2.16 | 10.06 | 0 | 18.01 |
| Richest | 5.12 | 0 | 6.05 | 9.55 | 0 | 7.14 | 10.97 | 28.88 | 4.63 |
| **Sex** | | | | | | | | | |
| Male | 6.25 | 6.39 | 6.65 | 9.43 | 8.35 | 4.04 | 13.87 | 15.22 | 12.86 |
| Female | 5.24 | 6.5 | 14.54 | 10.19 | 9.16 | 14.12 | 12.07 | 11.93 | 14.65 |
| **Age** | | | | | | | | | |
| Less than 6 months | 3.73 | 3.06 | 2.84 | 12.87 | 22.75 | 16.57 | 7.39 | 8.73 | 16.12 |
| 6–11 months | 8.88 | 9.03 | 14.09 | 20.58 | 14.59 | 18.93 | 20.14 | 16.61 | 23 |
| 12–23 months | 6.5 | 6.12 | 15.93 | 13.13 | 13.92 | 5.55 | 18.11 | 22.57 | 24.66 |
| 24–35 months | 5.01 | 8.19 | 15.71 | 8.33 | 5.72 | 8.41 | 12.62 | 12.71 | 14.1 |
| 36–47 months | 6.81 | 7.74 | 10.28 | 5.66 | 4.66 | 6.94 | 13.23 | 14.09 | 7.53 |
| 48–60 months | 4.63 | 4.67 | 5.03 | 5.43 | 1.16 | 6.05 | 8.71 | 7.61 | 9.43 |
| **Fathers' Working Status** | | | | | | | | | |
| Not Working | 3.71 | 4.55 | 0 | 11.34 | 7.23 | 2.15 | 13.81 | 7.05 | 1.62 |
| Working | 6.04 | 7.08 | 13.01 | 9.59 | 9.27 | 10.29 | 12.89 | 15.76 | 16.66 |
| **Mothers' Working Status** | | | | | | | | | |
| Not Working | 5.64 | 6.41 | 11.26 | 9.3 | 8.65 | 8.61 | 13.18 | 13.38 | 13.78 |
| Working | 6.48 | 8.33 | 0 | 12.75 | 14.68 | 7.42 | 11.94 | 24.1 | 12.04 |
| **Region** | | | | | | | | | |
| North | 4.86 | 4.79 | 5.41 | 9.33 | 9.15 | 7.73 | 13.16 | 13.48 | 21.51 |
| Central | 6.63 | 8.35 | 11.14 | 10.36 | 8.11 | 8.49 | 13.12 | 13.74 | 11.24 |
| South | 3.66 | 5.69 | 13.79 | 8.09 | 12.28 | 12.82 | 11.84 | 12.7 | 21.54 |
| **Total** | 5.76 | 6.45 | 10.13 | 9.8 | 8.76 | 8.49 | 13 | 13.57 | 13.66 |

Source: Authors' estimation using JPFHS (2017–18).

'Other-nationalist' & Jordanian, and children whose mothers with higher education of 'Syrian nationalists' are positively affected by fever. The highest prevalence of fever is found among Syrian and Jordanian male children than female. Children age 12–23 months have a high prevalence of fever. Children whose father is working have a higher fever prevalence than non-working fathers among both Syrian and Other-nationalist. Syrian children whose mother are working has the highest prevalence rate, with 24.1%. The southern region of Jordan has the highest prevalence, where the lowest is seen in the Central part among other nationalist children.

**Table 4.** Prevalence of acute respiratory infection, diarrhea and fever by demographic, health, hygiene and environmental characteristics among under-five children in Jordan, 2017–18.

| Characteristics | ARI | | | Diarrhea | | | Fever | | |
|---|---|---|---|---|---|---|---|---|---|
| | Jordanian | Syrian | Others | Jordanian | Syrian | Others | Jordanian | Syrian | Others |
| **Demographic factors** | | | | | | | | | |
| *Birth Order* | | | | | | | | | |
| <2 | 5.62 | 6.66 | 14.2 | 9.95 | 12.18 | 9.06 | 12.49 | 12.14 | 15.85 |
| 3–5 | 6.29 | 7.04 | 5.3 | 9.52 | 5.58 | 8.52 | 13.15 | 13.18 | 12.73 |
| 6+ | 3.82 | 4.37 | 3.48 | 10.41 | 7.87 | 3.42 | 15.11 | 18.36 | 2.19 |
| *Number of children living* | | | | | | | | | |
| Single | 6.3 | 5.35 | 24.65 | 15.07 | 22.67 | 10.95 | 14.88 | 15.16 | 18.58 |
| Three | 6.2 | 6.46 | 8.21 | 9.31 | 7.97 | 10.36 | 13.27 | 11.92 | 16.19 |
| More than three | 4.97 | 6.72 | 3.87 | 8.33 | 5.94 | 4.65 | 11.89 | 14.76 | 8.12 |
| **Health services** | | | | | | | | | |
| *Health card* | | | | | | | | | |
| No | 5.41 | 0.6 | 0 | 4.64 | 10.05 | 2.37 | 5.13 | 11.39 | 0 |
| Yes | 5.88 | 7.15 | 13.38 | 12.98 | 12.94 | 11.01 | 14.83 | 15.92 | 19.6 |
| *Received BCG* | | | | | | | | | |
| No | 4.64 | 1.51 | 0 | 10.87 | 11.84 | 0.78 | 9.54 | 14.49 | 12.51 |
| Yes | 5.99 | 7.45 | 14.74 | 12.91 | 12.88 | 12.12 | 15.04 | 15.79 | 19.83 |
| **Wash and hygiene** | | | | | | | | | |
| *Sources of water* | | | | | | | | | |
| Piped | 4.68 | 7.04 | 2.8 | 9.04 | 8.47 | 9.04 | 11.04 | 11.9 | 9.9 |
| Bottle | 6.66 | 6.41 | 23.26 | 10.06 | 9.68 | 10.06 | 14.91 | 16.74 | 20.25 |
| Others | 3.37 | 6.14 | 13.09 | 12.5 | 7.41 | 12.5 | 13.83 | 12.34 | 10.65 |
| *Plain Water* | | | | | | | | | |
| No | 3.93 | 2.63 | 1.34 | 9.81 | 5.95 | 2.86 | 6.44 | 5.05 | 7.91 |
| Yes | 6.31 | 6.19 | 12.73 | 11.91 | 12.77 | 11.17 | 14.52 | 15.63 | 21.53 |
| *Toilet Facilities* | | | | | | | | | |
| No | 5.38 | 6.3 | 9.5 | 9.61 | 8.34 | 7.56 | 12.66 | 13.09 | 13.35 |
| Yes | 8.82 | 10.67 | 14.9 | 10.02 | 14.29 | 31.6 | 18.48 | 22.69 | 18.72 |
| **Environment Factors** | | | | | | | | | |
| *Fuel type* | | | | | | | | | |
| Smokeless | 5.44 | 6.7 | 9.57 | 9.61 | 8.78 | 7.86 | 12.77 | 13.82 | 13.42 |
| Smoke | 0 | 5.77 | 0 | 15.3 | 31.09 | 0 | 0 | 43.74 | 0 |
| *Mothers' Cigarette* | | | | | | | | | |
| No | 5.75 | 6.44 | 10.94 | 9.82 | 8.84 | 7.07 | 13.17 | 13.51 | 14.74 |
| Yes | 5.91 | 6.62 | 0 | 9.42 | 7.53 | 26.23 | 10.57 | 14.58 | 0.97 |
| *Mother Smoking water* | | | | | | | | | |
| No | 5.5 | 6.63 | 8.89 | 9.51 | 8.87 | 8.04 | 12.82 | 13.72 | 13.9 |
| Yes | 10.18 | 0.69 | 29.7 | 14.54 | 5.39 | 15.58 | 15.92 | 8.94 | 10.19 |

Source: Authors' estimation using JPFHS (2017–18).

Table 4 presents the prevalence of acute respiratory infection, diarrhea, and fever by demographic, health, hygiene, and environmental characteristics among under-five children living in Jordan, 2017–18. The highest prevalence of ARI is observed among 'Other-nationalist' children with 14.2% and the lowest among Jordanians with birth order more than six. A single child is positively affected by ARI among Jordanian and Other-nationalist. Children who have

a health card and received the BCG vaccine are positively affected by ARI among all national-ists. 'Other-nationalist' children who consume bottled water have a high prevalence of ARI. Jordanian children whose mother has a smoking habit have high ARI prevalence compared to Syrian and Other nationalists.

However, the highest prevalence of diarrhea is observed among Syrian (12.18%) with birth order less than two, and the lowest is observed among 'Other-nationalist' (3.42%) with birth order more than six. Single Syrian children are positively affected by diarrhea, with 22.67%. Children who have health cards and received the BCG vaccine are positively affected by diar-rhea among all nationalists. Other sources of water supply among Jordanian and 'Other-nationalist' children have a high prevalence of diarrhea. Syrian children who have plain water supply are highly infected with diarrheal diseases. Around 31.6% of 'Other-nationalist' chil-dren are positively affected by the diarrheal disease, which has toilet facilities. The high preva-lence is observed among Syrian children, with approximately 31% who use smoke fuel. 'Other-nationalist' children whose mother has a smoking cigarette and smoking water habit have a high prevalence rate of diarrhea.

Similarly, the highest prevalence of fever is observed among Syrian (18.36%) whose birth order is more than six. Single children and children with health cards and received BCG are positively affected by fever in all nationalist groups. Other-nationalist children who consume water by bottles and have plain water supply have a high prevalence of fever. Syrian children who have toilet facilities are highly infected with fever. A high prevalence of fever is observed among Syrian children, with around 43.74% who use smoke fuel. Syrian and 'Other-national-ist' children whose mother has smoking cigarette habit are more likely to be affected by fever. Jordanian children whose mother has smoking water habits are positively affected by fever.

Table 5 presents multinomial logistic regression applied to determine the risk factors asso-ciated with under-five children living in Jordan (Jordanian, Syrian, and Other-nationalist) by its socioeconomic characteristics in 2017–18. 'Syrian nationalist' children have a higher rela-tive risk of ARI (RRR = 1.19, [1.08, 1.32]), and 'Other-nationalist' children have two times greater risk of ARI compared to 'Jordanian children.' The relative risk of diarrhea is lower among 'Syrian nationalist' and 'Other-nationalist' children compared to 'Jordanian children.' Children belong 'Other-nationalist' are found to be less relative risk of fever (RRR = 0.9, [0.80, 1.01]) than 'Jordanian children.'

Comparing to urban children, rural children's relative risk ratio (RRR) of 'Syrian national-ist' is (RRR = 0.16, [0.15, 0.18]) and 'Other-nationalist' (RRR = 0.26, [0.22, 0.31]) is less likely to be at risk. However, we found that the mother's education positively affects the nationalist group in Jordan. Though, the RRR among children whose mother has primary education is (RRR = 1.97, [1.73, 2.25]) for 'Syrian nationalist' and (RRR = 1.2, [0.97, 1.47]) for 'Other-nationalist' are higher compared to mother with no education. Similarly, children whose mother have secondary education (RRR = 0.29, [0.26, 0.33]) & higher education (RRR = 0.07, [0.06, 0.08]) for 'Syrian nationalist' and secondary education (RRR = 0.29, [0.26, 0.33]) & higher education (RRR = 0.21, [0.18, 0.25]) for 'Other-nationalist' have lesser relative risk com-pared to mother with no education. Compared to the poorest children, all wealth index catego-ries have lower relative risk ration for 'Syrian nationalist' children. At the same time, the richest children who belong to the 'Other-nationalist' have higher relative-risk (RRR = 1.16, [1.03, 1.3]) compared to the poorest children. Further, the result shows that 'Syrian nationalist' children have four times more relative risk residing in the Southern region and less relative risk (RRR = 0.52, [0.51, 0.56]) residing in the Central region than compared to the Northern region of Jordan. Despite that, the RRR among 'Other-nationalist' children is (RRR = 1.12, [1.06, 1.19]) for Central region and (RRR = 0.2, [0.16, 0.24]) for Southern region compared to the Northern region of Jordan.

**Table 5. Multinomial regression results of nationalities showing relative risk ratio among under-five children in Jordan (2017–18) by its diseases and socioeconomic characteristics.**

| Diseases and Socioeconomic characteristics | Syrian | | | Other-Nationalist | | |
|---|---|---|---|---|---|---|
| | RRR | Confidence Interval | | RRR | Confidence Interval | |
| | | Lower | Upper | | Lower | Upper |
| **ARI** | | | | | | |
| NoⓇ | | | | | | |
| Yes | 1.19*** | 1.08 | 1.32 | 2.16*** | 1.90 | 2.45 |
| **Diarrhea** | | | | | | |
| NoⓇ | | | | | | |
| Yes | 0.77*** | 0.71 | 0.84 | 0.87** | 0.76 | 0.98 |
| **Fever** | | | | | | |
| NoⓇ | | | | | | |
| Yes | 1.00 | 0.93 | 1.08 | 0.90* | 0.80 | 1.01 |
| **Place of Residence** | | | | | | |
| UrbanⓇ | | | | | | |
| Rural | 0.16*** | 0.15 | 0.18 | 0.26*** | 0.22 | 0.31 |
| **Mothers' Education** | | | | | | |
| No EducationⓇ | | | | | | |
| Primary | 1.97*** | 1.73 | 2.25 | 1.20* | 0.97 | 1.47 |
| Secondary | 0.29*** | 0.26 | 0.33 | 0.22*** | 0.18 | 0.27 |
| Higher | 0.07*** | 0.06 | 0.08 | 0.21*** | 0.17 | 0.25 |
| **Wealth Index** | | | | | | |
| PoorestⓇ | | | | | | |
| Poorer | 0.32*** | 0.30 | 0.33 | 0.90** | 0.82 | 0.99 |
| Middle | 0.12*** | 0.11 | 0.13 | 0.45*** | 0.40 | 0.50 |
| Richer | 0.10*** | 0.09 | 0.11 | 0.50*** | 0.44 | 0.56 |
| Richest | 0.10*** | 0.09 | 0.12 | 1.16** | 1.03 | 1.30 |
| **Regions** | | | | | | |
| NorthⓇ | | | | | | |
| Central | 0.53*** | 0.51 | 0.56 | 1.12*** | 1.06 | 1.19 |
| South | 4.29*** | 3.74 | 4.93 | 0.20*** | 0.16 | 0.24 |

***p<0.01,

**p<0.05,

*p<0.1;

Ⓡ-Reference category;

ARI- acute respiratory infection.

Source: Authors' estimation using JPFHS (2017–18).

## Discussion

The aim of this paper was to investigate the impact of conflict due to displacement on health care outcomes among children under-five years of age. We examined the health outcomes for children of mothers belonging to various nationalities in Jordan. Given that Jordan's health system faces an inevitable challenge in meeting the unpredictable demand on healthcare services due refugee influx from its neighboring countries [44], we examined set of health outcomes namely: diarrhea, ARI and fever among under-five children in Jordan based on their mother's nationality.

We found that health outcomes are asymmetric among the children of different nationalities, with children from Jordan performing better in outcomes like ARI and fever. This may be due to Jordanian mothers have better access to healthcare facilities as compared to mothers of other nationalities displaced due to conflict which may affect the health outcomes of their children. These findings are consistent with the earlier studies that reported that displaced children are at greater risk in terms of health outcomes and provision of health care services [45]. Despite the fact that Jordan has one of the most modern healthcare infrastructures in the Middle East countries [46], children mostly coming into Jordan due to conflict in their own countries face the challenges of access to health care services and are associated with more significant health risks as shown by our results. This has been also witnessed in earlier studies, where children from other nationalities were found to be at greater risk than the Jordanian children [47]. One other study revealed that the Jordan region's political situation associated with the war made it more challenging to study child health issues, especially among the refugee populations who have minimal information about child health and are more vulnerable to health risks [48].

While examining the health risks associated with children belonging to different nationalities in Jordan, we found an ample evidence of health inequality among the children with Syrian and other nationalities, particularly in ARI and Fever. Though the diarrhea was marginally higher among the Jordanian children due to poor household sanitary conditions and adequacy of water facilities, especially among the poor households in Jordan's rural areas Furthermore, Jordan is one of the scariest countries in terms of freshwater, which increases the likelihood of poor sanitation and increases diarrheal diseases risk [49]. However, given the large number of refugees in urban areas' sanitation programs and sewage networks have been implemented among the refugees' concentrated camps over time, which might reduce diarrheal risk among the children of other nationalities [50].

The increasing risk of health outcomes like ARI and fever among children has much to do with individuals' socio-economic levels and well-being [51]. People coming through displacement mainly have no or low access to healthcare and lack basic health care facilities. Apart from that, they are at greater risk of contracting the disease outbreaks due to poor amenities and living arrangements as found in previous literature [52,53]. Another possible reason for the lack of access to healthcare services is being not registered with the local health care system to receive benefits. Our results suggest that, children belonging to these nationalities are at a greater risk both in terms of exposure to diseases and receiving the benefits from local health care services.

Prior studies in this filed reveal that more significant inequities are well reflected in access to services among the Syrian and other nationalities than the Jordanian children [54]. This study also found the same while using mother's nationality as a proximity for displaced people in Jordan using the data from JPFHS. Thus, additional studies are needed to identify the health care challenges among children in Jordan especially who belong to Syrian and other nationalities. Similarly understanding the health risks and other possible challenges associated with children belonging to Syrian and Other nationalities living in Jordan, through some primary surveys can be also significant to understand the health care inequality among this group. Conflict negatively impacts health outcomes. Though public health research in terms of conflict is limited due to multiple challenges including consistent life threats, security concerns, and movement restriction, apart from the fragile health care system and lack of sufficient data in this context. Importantly, our findings emphasize that armed conflict is a public health problem, which has serious impact on vulnerable groups like children. Children exposed to any conflict or any other difficult situation are likely susceptible to health risks apart from the risk of exploitation and sexual abuse [55,56]. Therefore, significant attention needs to be put forth

through policy measures to address these public health challenges, which may have potential impact on their health and wellbeing.

## Limitations

This study is the first kind in an approach to link the conflict-driven displacement with child health outcomes based on secondary data. Although the sample size was significant enough, linking it with conflict was not explicit. There was no direct evidence of mothers belonging to different nationalities affected by conflict-driven displacement. This study provides useful insights into health outcomes facing vulnerable populations, but being cross-sectional does not imply the causal inference. The study further could not examine the mental health outcome, which is one of the children's critical issues due to conflict.

## Conclusions

The conflict has inevitable effects on the health and loss of life. Conflict puts vulnerable populations like children at greater risk through preventable diseases like diarrhea, ARI, and other chronic diseases as found in our study. Conflict-driven displacement has an immediate effect on child health through access, availability, and affordability of health care services, resulting in more significant health care risks. This likely risks the health and wellbeing of populations at risk especially among children, women and elderly. This study found a significant impact of displacement on the poor health outcomes among children like ARI and fever. Therefore, concerted action is required to safeguard the health needs and avert public health emergencies due to conflict driven displacement. Coordinated and effective measures are needed to provide the best health care services among the displaced populations to prevent health risks.

Moreover, investing in health is investing in peace; therefore, sufficient investment is required to address such adversities that affect the health care system due to uneven demand as experienced by the Jordanian health care system. Thus, collaborative efforts through global partners can play a significant role in the countries facing the challenges of managing these health care emergencies. To sum up, studies like this can also provide some useful insights into understanding how health strategies can be implied in this context to avoid adverse health outcomes and provide an inclusive health care approach for all health benefits.

## Implications

There is a compelling need for research in this domain to be extended to understand the impact of conflict and displacement on child health and development. Although this study provides some basis for conducting such research based on secondary data, there is a dearth for future research in this context to address the public health issues of children and their mothers. Some of the key recommendations for both policymakers and researchers to address based on the linkages examined above in the study are as follow:

**Practical implications.** Researchers must examine the social determinants of physical health in conflict settings, which have a likely impact on children's physical health and wellbeing. There is a need to design specific surveys in order to understand the problems at greater depth so that strategies adopted can be empirically tested. Effective intervention strategies should also be designed to prevent the avoidable mortality and morbidities affecting the children due to conflict-driven displacement. Designing appropriate health promotion programs is also worthwhile to enhance awareness regarding these diseases.

**Policy implications.** It is important to ensure that health care service provisions are accessible to all. There is a need to strengthen the coverage of health care services, emergency care services and ensure mobile care services among displaced populations. Health care assistance

to the Jordanian government faces unprecedented demand and investment grants in health care services.

## Supporting information

**S1 Table. Additional information of the variables that have been categorized in the data analysis.**
(DOCX)

## Author Contributions

**Conceptualization:** Manzoor Ahmad Malik.

**Data curation:** Manzoor Ahmad Malik.

**Formal analysis:** Manzoor Ahmad Malik, Saddaf Naaz Akhtar.

**Methodology:** Saddaf Naaz Akhtar.

**Supervision:** Manzoor Ahmad Malik.

**Writing – original draft:** Saddaf Naaz Akhtar.

**Writing – review & editing:** Manzoor Ahmad Malik, Saddaf Naaz Akhtar, Rania Ali Albsoul, Muhammad Ahmed Alshyyab.

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
