## [Decision Letter · Decision Letter 0]

2 Mar 2021

PONE-D-20-38398

Conflict driven displacement and child health: Evidence from Jordan

PLOS ONE

Dear Dr. Akhtar,

Thank you for submitting your manuscript to PLOS ONE. After careful consideration, we feel that it has merit but does not fully meet PLOS ONE’s publication criteria as it currently stands. Therefore, we invite you to submit a revised version of the manuscript that addresses the points raised during the review process.

Considering reviewers comments, I am going with a decision of Major revision 

We look forward to receiving your revised manuscript.

Kind regards,

Srinivas Goli, Ph.D.

Academic Editor

PLOS ONE

Journal Requirements:

2. In your Methods section, please provide additional information about the eligibility criteria applied in your analysis; and please clarify how every variable was categorised.

3. Please amend your manuscript to include your abstract after the title page.

Additional Editor Comments:

Considering reviewers comments, I am going with a decision of Major revision

Reviewers' comments:

Reviewer's Responses to Questions

**Comments to the Author**

1. Is the manuscript technically sound, and do the data support the conclusions?

Reviewer #1: Yes

Reviewer #2: Partly

Reviewer #3: Yes

2. Has the statistical analysis been performed appropriately and rigorously? 

Reviewer #1: I Don't Know

Reviewer #2: Yes

Reviewer #3: Yes

3. Have the authors made all data underlying the findings in their manuscript fully available?

Reviewer #1: Yes

Reviewer #2: Yes

Reviewer #3: Yes

4. Is the manuscript presented in an intelligible fashion and written in standard English?

Reviewer #1: Yes

Reviewer #2: Yes

Reviewer #3: Yes

5. Review Comments to the Author

Reviewer #1: Useful literature can be taken up in the discussion section. These papers discuss the risk factors for unexpected child death, sudden infant death and stillbirth in Jordan and the many issues raised regarding the health care system, high smoking rate and smoking restrictions, as well as cultural, economic and risk groups.

-Hamadneh S., Kassab M., Eaton A., Wilkinson A., Creedy D.K. (2020) Sudden Unexpected Infant Death. In: Laher I. (eds) Handbook of Healthcare in the Arab World. Springer, Cham. https://doi.org/10.1007/978-3-319-74365-3_144-1

- Hamadneh S, Kassab M, Hamadneh J, Amarin Z. Sudden unexpected infant death in Jordan and the home environment. Pediat Int. 2016; 58(12):1333-1336.

- Khader Y, Batieha A, Khader A, Hamadneh S. (2018) Stillbirths in Jordan: rate, causes, and preventability, The Journal of Maternal-Fetal & Neonatal Medicine, Published online: 25 Sep 2018. DOI: 10.1080/14767058.2018.1517326

- Hamadneh S, Al-Shdayfat N, Al-Omari O, Hamadneh J, Bashtawi M, Alkhatib A, Amarneh B, Andre K, Willkinson A. Sudden Infant Death Syndrome in the Middle East: An Exploration of the Literature on Rates, Risk Factors, High Risk Groups and Intervention Programs. Research Journal of Medical Sciences. 2016;10(4):199-204

- Hamadneh S. Sudden unexpected infant deaths investigation in the Middle East requiring further action· GSTF Journal of Nursing and Health Care (JNHC). 2016;4(1): 53-56

Reviewer #2: The article is written in a clear and understandable language. The data is described as it is. However, (1) the discussion contradicts the result on diarrhea. The result shows that prevalence of diarrhea is higher in Jordanian children than Syrians and other nationalities. (2) The difference in health outcomes seems correlated to socio-economic indictors than nationality. The authors should focus the article on the socio-economic disparity between refugees and Jordanians affecting health of children. The nationality argument not supported well. (3) How does the ARI, diarrhea and fever data look like by nationality, when controlled for socio-economic variables? (4) What do the authors think is the reason for prevalence of diarrhea being higher in Jordanian children than in Syrians or other nationalities? I suggest including that routine immunization is free for all children under 5 in Jordan. However, pneumococcal conjugate vaccine (PCV) is not part of Jordan's routine immunization program. When it is introduced, it will be available for all children under 5, including refugees.

Reviewer #3: 1. It is a very important study on the health of the children under five having different nationalities, all residents of Jordan.

2. Please modify the title according to the objective of the study.

3. The objective of the study needs to be precisely restructured and focused.

4. The Abstract has to be written properly and concisely.

5. The Conclusion of the study has to be consistent with the study findings.

6. In Table 2, sample size has to be recalculated.

7. Please take out the wordings like PhD Scholar, only the PhD will suffice. What about the credentials of other coauthors, please mention.

8. The limitations of the study are nicely narrated.

9. Please make necessary corrections of grammar and restructuring the sentences.

6. PLOS authors have the option to publish the peer review history of their article (what does this mean?). If published, this will include your full peer review and any attached files.

Reviewer #1: **Yes: **Shereen Hamadneh

Reviewer #2: No

Reviewer #3: No

---

## [Author Response · Author response to Decision Letter 0]

10 Apr 2021

Title: Conflict driven displacement and child health inequality: Evidence based on mother’s nationality from Jordan Population and Family Health Survey 

Dear All,

We would like to thank you for these constructive, building and improvable comments on this manuscript that would improve the manuscript's substance and content. We considered each comments and clarification questions of the editor and reviewers on the manuscript thoroughly. Our point-by-point responses for each comment and questions are described in detailed on the following pages. Further, the details of changes were shown by track changes in the supplementary document attached. Once again, thank you for your valuable insights.

Editor’s comments

 Authors’ response: Dear Editor, Thank you for the suggestion. We have prepared the manuscript as per the PLOSONE’s style including file name. 

2. In your Methods section, please provide additional information about the eligibility criteria applied in your analysis; and please clarify how every variable was categorised.

Authors’ response: Thank you. We have prepared the additional information about the outcome and explanatory variables used in the data analysis. This can be found in S1 Table. Kindly see the supporting information’s file.

3. Please amend your manuscript to include your abstract after the title page.

Authors’ response: Thank you for your valuable suggestion. We have included the abstract after the title page.

Reviewer #1

Comment: Useful literature can be taken up in the discussion section. These papers discuss the risk factors for unexpected child death, sudden infant death and stillbirth in Jordan and the many issues raised regarding the health care system, high smoking rate and smoking restrictions, as well as cultural, economic and risk groups.

Authors’ response: We have included literature as suggested by the reviewer in the discussion section. Thank you for the informative suggestion.

Reviewer #2

1. The discussion contradicts the result on diarrhea. The result shows that prevalence of diarrhea is higher in Jordanian children than Syrians and other nationalities. 

Authors’ response: Thank you for your comments we have critically revised our discussion and made it consistent with the study findings in the revised draft, we kindly suggest you to go through the revised draft.

2. The difference in health outcomes seems correlated to socio-economic indictors than nationality. The authors should focus the article on the socio-economic disparity between refugees and Jordanians affecting health of children. The nationality argument not supported well. 

Authors’ response: Thank you for highlighting the core issue of this paper. As we framed this study, we also have a similar concern, but our aim was to examine the possible health risk due to conflict-driven displacement despite having a good health system as in the case of Jordan, which has one of the best health care system accommodating millions of refugees in the world. So, we included nationality as an important factor in our study. 

Secondly, since this was a study based on secondary data, without any direct evidence of the mother’s being likely from Syria. So, we derived a notion based on some earlier literature and theoretical construct to establish the link between conflict and child health outcomes via mother’s nationality. 

Thirdly health outcomes had a strong correlation with the children’s nationality derived from their mother’s. However, socio-economic indicators were also correlated with child health outcomes, but since our theoretical notion was more about displacement, so that was the best proxy indicator available in the dataset to be undertaken. Furthermore, socio-economic disparities have been already examined as cited by our study, so we used nationality as a construct to study health outcomes. 

Lastly, although, we could not find sufficient evidence to support the nationalities argument, except denoting it as proxy mainly derived from conflict-driven displacement. But we did mention it as a possible limitation in our study.

3. How does the ARI, diarrhea and fever data look like by nationality, when controlled for socio-economic variables? 

Authors’ response: Thank you. The results were similar even after controlling them, we have already taken this into account while carrying our initial analysis, but since our goal was to study nationality as a likely factor based on the notion of conflict-driven displacement, so we used multinomial modelling to accommodate it, rather than control it as a potential predictor in the final analysis.

4. I suggest including that routine immunization is free for all children under 5 in Jordan. However, the pneumococcal conjugate vaccine (PCV) is not part of Jordan's routine immunization program. When it is introduced, it will be available for all children under 5, including refugees.

Authors’ response: Thank you for your suggestion. We have taken note of it in our revised manuscript.

Reviewer #3

1. It is a very important study on the health of the children under five having different nationalities, all residents of Jordan.

Authors’ response: Yes. It is.

2. Please modify the title according to the objective of the study.

Authors’ response: Since the focus was more on establishing a connotation of conflict with health outcomes, we felt it more catchy, but we have modified the title and made it more precise and problem-oriented.

3. The objective of the study needs to be precisely restructured and focused.

Authors’ response: Thank you. We have restructured it and made it more relevant with title and theoretical construct.

4. The Abstract has to be written properly and concisely.

Authors’ response: Thank you for the suggestion. The abstract has been concisely written. We would like you to see the revised manuscript.

 5. The Conclusion of the study has to be consistent with the study findings.

Authors’ response: Thank you for your comment. We have revised the conclusion and made it consistent with the study findings. 

6. In Table 2, sample size has to be recalculated.

Authors’ response: Thank you for your comment, we have recalculated the Table and attached the same in the updated version. We kindly suggest to see the revised supplementary document.

Nationality ARI Diarrhea Fever Total Sample*

Jordanian 5.76 9.8 13.01 8065

Syrian 6.45 8.76 13.57 1191

Other 10.13 8.49 13.01 368

Total 6.02 9.62 13.09 9623

*Unweighted total sample of Children by mother’s nationality

Source: Jordan Demographic and Health Surveys 2017-18.

7. Please take out the wordings like PhD Scholar, only the PhD will suffice. What about the credentials of other coauthors, please mention.

Authors’ response: Dear Reviewer, I have updated my personal details concisely. But the online submission system didn’t show the updated version. 

And I am not able to edit the other coauthors credential because I am not getting such options to edit in the online submission system. But I have mentioned in the cover letter. Kindly see the cover letter.

8. The limitations of the study are nicely narrated.

Authors’ response: Thank you.

9. Please make necessary corrections of grammar and restructuring the sentences.

Authors’ response: Thank you for your kind suggestions. The necessary corrections of grammar and reconstruction of the sentences have been made clearly. We kindly request you to kindly see the revised manuscript.

---

## [Decision Letter · Decision Letter 1]

18 Jun 2021

PONE-D-20-38398R1

Conflict driven displacement and child health inequality: Evidence based on mother’s nationality from Jordan Population and Family Health Survey

PLOS ONE

Dear Dr. Akhtar,

Thank you for submitting your manuscript to PLOS ONE. After careful consideration, we feel that it has merit but does not fully meet PLOS ONE’s publication criteria as it currently stands. Therefore, we invite you to submit a revised version of the manuscript that addresses the points raised during the review process.

ACADEMIC EDITOR: Reviewer 2 is still not happy with writing style of the manuscript, thus I am sending it back to you. Can you carefully give attention to reviewer 2 comments. 

We look forward to receiving your revised manuscript.

Kind regards,

Srinivas Goli, Ph.D.

Academic Editor

PLOS ONE

Additional Editor Comments (if provided):

Reviewer 2 is still not happy with writing style of the manuscript, thus I am sending it back to you. Can you carefully give attention to reviewer 2 comments.

Reviewers' comments:

Reviewer's Responses to Questions

**Comments to the Author**

1. If the authors have adequately addressed your comments raised in a previous round of review and you feel that this manuscript is now acceptable for publication, you may indicate that here to bypass the “Comments to the Author” section, enter your conflict of interest statement in the “Confidential to Editor” section, and submit your "Accept" recommendation.

Reviewer #1: All comments have been addressed

Reviewer #2: (No Response)

2. Is the manuscript technically sound, and do the data support the conclusions?

Reviewer #1: Yes

Reviewer #2: No

3. Has the statistical analysis been performed appropriately and rigorously? 

Reviewer #1: I Don't Know

Reviewer #2: Yes

4. Have the authors made all data underlying the findings in their manuscript fully available?

Reviewer #1: Yes

Reviewer #2: Yes

5. Is the manuscript presented in an intelligible fashion and written in standard English?

Reviewer #1: Yes

Reviewer #2: No

6. Review Comments to the Author

Reviewer #1: It is possible to start the process of publishing. The researchers made the modifications identified in the review.

Reviewer #2: The topic of study is still relevant and important. However, I was not able to discern the man public health message from the data and discussions. The revised manuscript is even more confusing than the previous version. The results are not described and discussed in a meaningful and coherent way that makes sense in terms of their public health relevance. I could not understand, what the authors were trying to tell us. For instance,

"..results showed that Jordanian children whose mother has a smoking habit and Syrian children's mothers who do not smoke cigarettes have high ARI prevalence. Hence, our results clearly show that disease risk is greater

among the children of Syrian or any other nationality than the Jordanian."

The authors should have picked a few of the results, which they thought were most relevant to support the title of the article. In its current form, I do not see this article contains coherent and, from public health perspective, relevant new information that merits publication in PLOS.

7. PLOS authors have the option to publish the peer review history of their article (what does this mean?). If published, this will include your full peer review and any attached files.

Reviewer #1: **Yes: **Shereen Hamadneh

Reviewer #2: No

---

## [Author Response · Author response to Decision Letter 1]

20 Aug 2021

Thank you for your critical evaluation. We have revised once again our paper and made the changes as highlighted by you. We have tried our best this time to make the results coherent with the final objective and title of the research paper.

Now I would like to come to your specific comments 

Regarding the public health message of the paper. 

While conceptualizing the paper, we tried to address some of the key aspects that are key in terms from public health perspective only. So, our aim was to make a comprehensive analysis of how health care system of a particular country gets affected by uncertain demands that arise mainly from the conflict in its neighbouring states. Whether the influx of refugees overshadows the health care services for natives. And with risk of vulnerability in health care outcomes how does these public health challenges impact the health of refugee’s especially vulnerable population like children. Therefore, based on the children sample determined by mother’s nationality we used the JDHS to analyze the health outcomes among the under-five children in Jordan including native Jordanian children and other nationality children.

We felt that the paper critically analyses the impact of conflict in health settings particularly in terms of provision of healthcare services and inequality in terms of access to health care. We therefore felt this an in important significant public health issues undertaken by the PLOSone in earlier research, that is why we opted for this journal, and felt that it will reach the wider audience given its strong implications in terms of health care access and provisions in children.

Regarding your results suggestion

Thank you for highlighting that we have already corrected it in our final revision. 

I have hoped this revision of paper suits well and address the concerns, which you have pointed out. We will be really pleased if there is any other issue that we need to address in this paper. We would love to correct it before making it suitable for this publication.

Regards 

Authors

---

## [Editor Report · Decision Letter 2]

24 Aug 2021

Conflict driven displacement and child health inequality: Evidence based on mother’s nationality from Jordan Population and Family Health Survey

PONE-D-20-38398R2

Dear Dr. Akhtar,

We’re pleased to inform you that your manuscript has been judged scientifically suitable for publication and will be formally accepted for publication once it meets all outstanding technical requirements.

Kind regards,

Srinivas Goli, Ph.D.

Academic Editor

PLOS ONE

Additional Editor Comments (optional):

Authors have implemented the reviewers concerns. Thus, I am recommending this paper with minor suggestions.

1. Remove the word "inequality" from the paper title, as you really not estimating any inequality in the paper.

2. Under each table and figures authors wrote source is JDHS, please replace it as "Authors estimation using JDHS".

3. Replace the recommendation section heading as "Implications". Also replace sub-heading "for researchers" with "Practical implications", and "For Government" with "Policy implications". And, write this two sections as a coherent paragraph rather than bullet points.
---

## [Editor Report · Acceptance letter]

26 Aug 2021

PONE-D-20-38398R2 

Conflict driven displacement and child health: Evidence based on mother’s nationality from Jordan Population and Family Health Survey 

Dear Dr. Akhtar:

I'm pleased to inform you that your manuscript has been deemed suitable for publication in PLOS ONE. Congratulations! Your manuscript is now with our production department. 

Kind regards, 

on behalf of

Dr. Srinivas Goli 

Academic Editor

PLOS ONE